# Achieving High Expression of Cry in Green Tissues and Negligible Expression in Endosperm Simultaneously via *rbcS* Gene Fusion Strategy in Rice

**DOI:** 10.3390/ijms24109045

**Published:** 2023-05-20

**Authors:** Hao Chen, Yuqing Huang, Mengnan Ye, Ya Wang, Xiuying He, Jumin Tu

**Affiliations:** 1Institute of Crop Science, College of Agriculture and Biotechnology, Zhejiang University, Hangzhou 310058, China; isco@zju.edu.cn (Y.H.); yemengnan@zju.edu.cn (M.Y.); wangya840212@163.com (Y.W.); 2Rice Research Institute, Guangdong Academy of Agricultural Sciences, Guangzhou 510640, China; hexiuying@gdaas.cn; 3Guangdong Key Laboratory of New Technology in Rice Breeding, Guangzhou 510640, China; 4Key Laboratory of Genetics and Breeding of High Quality Rice in Southern China (Co-Construction by Ministry and Province), Ministry of Agriculture and Rural Affairs, Guangzhou 510640, China

**Keywords:** *OsrbcS*, *Cry1Ab/Cry1Ac*, fuse, insect resistant, transgenic rice

## Abstract

To allay excessive public concern about the safety of transgenic foods, and to optimize insect-resistant genes expression to delay the evolution of resistance in pests, we developed a promising strategy to fuse the GOI (gene of interest) with *OsrbcS* (rice small subunit of ribulose bisphosphate carboxylase/oxygenase) in transgenic rice, which acted as a carrier, driven by the *OsrbcS* native promoter to sequester its expression in green tissues. Using *eYFP* as a trial, we reported a high-level accumulation of eYFP in green tissue and almost none in the seed and root of the fused construct compared to the non-fused construct. After applying this fusion strategy in insect-resistant rice breeding, recombinant *OsrbcS*-*Cry1Ab/Cry1Ac* expressed rice plants conferred high resistance to leaffolders and striped stem borers, among which two single-copy lines possessed normal agronomic performance in the field. Specifically, Cry1Ab/Cry1Ac protein levels in single-copy construct transgenic lines ranged from 1.8 to 11.5 µg g^−1^ in the leaf, higher than the *Actin I* promoter-driven control, T51-1, about 1.78 µg g^−1^ in the leaf, but negligible (only 0.00012–0.00117 µg g^−1^) in endosperm by ELISA analysis. Our study provided a novel approach to creating Cry1Ab/Cry1Ac-free endosperm rice with a high level of insect-resistant protein in green tissues through the simultaneous usage of the *OsrbcS* promoter and *OsrbcS* as a fusion partner.

## 1. Introduction

Insect pests present a major threat to crop productivity and cause annual losses in the order of ten million tonnes [1]. Rice, as the staple food crop that feeds half the world’s population [2], is subject to three major detrimental lepidopteran insects throughout the entire period of plant growth: yellow stem borer (*Tryporyza incertulas* Walker), striped stem borer (*Chilo suppressalis* Walker), and leaffolder (*Cnaphalocrocis medinalis* Guenec) [3]. Compared to the use of chemical insecticides, biological control using biopesticides derived from *Bacillus thuringiensis*, such as crystal toxins (Cry) [4], cytolytic toxins (Cyt) [5], vegetative insecticidal proteins (Vips) [6], and secreted insecticidal protein (Sip) [7], is a more environment-friendly and efficient alternative for the control of insect pests. In particular, the Bt proteins in genetically engineered crops are not toxic to people, other vertebrates, or most beneficial insects. The field performance of genetically modified (GM) Bt crops in pest suppression ensures agricultural productivity and farm income economically. These GM Bt crops, instead of chemical insecticides, avoid environmental pollution and damage to the ecosystem from an agroecological perspective [8,9].

Although great economic, social, and ecological benefits have been realized with the adoption of GM Bt crops worldwide, this experience has not facilitated the adoption of GM Bt crops as a staple food because of excessive concern over GM food safety [10]. For instance, Huahui No.1 (T51-1 line) and Bt shanyou63, two famous Bt rice lines [11], were issued biosafety certificates in 2009, and Huahui No.1 was even approved for consumption by the US Food and Drug Administration in 2018, having not been previously allowed to enter the Chinese agricultural system. One of the main reasons for this is the public acceptance and concern about GM food safety [12], even though multiple research results have suggested that both Bt rice products are as safe as non-Bt control rice products as food [13]. 

Another problem with Bt crops is that the evolution of practical resistance by some pests has reduced the efficacy of Bt crops [14]. Practical resistance to Cry toxins in Bt crops, which is a field-evolved resistance that has practical consequences for pest management, has been documented in at least 26 cases involving 11 pest species in 7 countries [15]. For instance, Bt cotton has been effective against pink bollworm in China for over 20 years. However, the efficacy of Bt crops is reduced when pests evolve resistance, as exemplified by pink bollworm resistance to Bt cotton in India [8]. The high-level expression of Bt proteins is an effective strategy in delaying pest resistance to Bt crops, unlike other strategies, such as abundant refuges, pyramids of toxins, novel toxins, and so on [16,17].

Considering the metabolic burden imposed on plants, energy consumption, and a residual amount of insect-resistant protein in the edible parts of plants resulting from constitutive promoters, green tissue specific promoters, such as the *rbcS* promoter, are a better choice in plant genetic engineering. To date, the *rbcS* promoter has been successfully exploited to drive different resistant genes specific in green tissues and minimize the accumulation in seeds in engineering rice [18,19,20], and Bt-free rice endosperm especially may improve public acceptance and reduce concern about GM food safety [21,22,23].

Meanwhile, unfavorable expression levels are often encountered when introducing foreign genes into the host. This obstacle can be overcome by endogenous gene fusion to the foreign target [24,25]. For example, Takaiwa et al. utilized the seed storage protein, 16 kDa Cys-rich prolamins, as a fusion carrier to produce small size bioactive peptides in ER-derived protein bodies, thus yielding as high as 200 µg/grain recombinant protein in transgenic rice seeds [26]. Another successful application was reported in the cyanobacteria that the endogenous CpcB (cyanobacteria phycocyanin β-subunit) was translationally N-terminal placed to PHLS (β-phellandrene synthase) to get substantially higher expression than the PHLS protein alone under the same endogenous *cpc* promoter. The CpcB-PHLS fusion protein retained the activity of the PHLS enzyme and catalyzed β-phellandrene synthesis reaction, yielding an average of 3.2 mg product g^−1^ dry cell weight (dcw), a 100-fold yield improvement compared to previous reports [27]. 

In this study, we took a strategy to fuse the GOI with *OsrbcS*, which acted as a carrier under the control of its native promoter to optimize the expression of the GOI in green tissues. Validation was conducted using *eYFP* as a trial, and there was no signal in the endosperm in RRY (pOsrbcS:rbcS-eYFP for *OsrbcS* fused) transformants, but a comparable level in the green tissue between RRY and RY (pOsrbcS:eYFP for *OsrbcS* unfused). Moreover, the rbcS-Cry1Ab/Cry1Ac expressing rice conferred strong resistance to leaffolders and striped stem borers in field experiments. ELISA analysis showed a high amount of Cry1Ab/Cry1Ac in the leaves to delay the pests’ resistance, while almost none existed in the endosperm, which could allay public concern about the safety of GM foods. Therefore, our study provided an enlightening approach to meet biosafety standards and tackle the evolution of pest resistance in insect-resistant rice breeding.

## 2. Results

### 2.1. Comparison of OsrbcS Fused and OsrbcS Unfused Transgenic Rice Plants

To confine the expression of the GOI strictly in the pest-attacked tissues without any reduction in the amount, we prepared two constructs using *eYFP* as a trial: pOsrbcS:eYFP for OsrbcS unfused (namely RY) and pOsrbcS:rbcS-eYFP for OsrbcS fused (namely RRY), under the control of the rice green-specific promoter *pOsrbcS* (Figure 1). For each construct, we randomly chose three transgenic lines for further analysis. As shown in Figure 2, the *eYFP* expression level was determined by qRT-PCR in four different tissues: leaf, stem, root, and seeds. In the green tissues, leaf and stem, the levels of *eYFP* in RRY and RY transformed lines were comparable (Figure 2A,B). Compared to the *CaMV 35S* promoters, the RRY plants were just one-fold lower in leaf tissue. However, in the non-green tissues, root and seed, the levels of the *eYFP* transcripts in the RRY plants were lower than those in the RY plants (Figure 2C,D). In seeds especially, the levels of the eYFP transcripts in the RRY plants were 1/13–1/7 to the RY plants. 

Similar results were observed under confocal laser scanning analysis. As shown in Figure 3, the positive control line p35S:eYFP exhibited high expression in all tissues (leaf, root, and seed), while the RY plants showed stronger signals in the leaf, but only weak signals in the root and the aleurone cells of seeds (Figure 3 arrow). The RRY plants showed similar fluorescence intensity compared to the RY plants in the leaf tissues, but were barely detected in the roots and seeds, similar to the negative controls, which contained no eYFP protein.

### 2.2. Generation and Protein Quantification of RRC Lines

According to the above, using *OsrbcS* as a fusion partner presents high expression in green tissue and reduced expression in seed organs. The validation trial results satisfied the requirements of null expression in endosperm in insect-resistant rice breeding. Therefore, we took this fusion strategy in insect-resistant breeding using the fusion gene *Cry1Ab/Cry1Ac*, which was characterized before [11]. Transgenic lines (namely RRC) were generated using the EHA105 Agrobacterium-mediated transformation method, and eight RRC lines without significant phenotypic changes were selected based on visual observation in the T_0_ generation. The copy number of T-DNA insertion in the transgenic plants was determined by Southern blot analysis. As shown in Figure 4A, there were five single-copy lines. The Cry1Ab/Cry1Ac protein concentrations of these five single-copy lines ranged from 2.54 µg g^−1^ leaf fresh weight in RRC-12 to 10.19 µg g^−1^ in RRC-11 at the tillering stage (Figure 4B), and ranged from 1.80 µg g^−1^ leaf fresh weight in RRC12 to 11.56 µg g^−1^ in RRC-1 at the filling stage (Figure 4B). All lines possessed higher protein levels compared to the elite transgenic insect-resistant rice T51-1 [11], in which the Cry1Ab/Cry1Ac gene was driven by the constitutively active rice *Actin I* promoter [28] in the leaf, both at the tillering and filling stage (Figure 4B). In contrast, the protein amount in the stem is around 1 µg g^−1^ at the tillering and filling stage in both T51-1 and RRC lines (Figure 4C). However, the Cry1Ab/Cry1Ac protein level was nearly zero in the endosperm. The minimum concentration was only 0.00012 µg g^−1^ in the RRC-12 line. Even the highest concentration was only 0.00117 µg g^−1^ in the RRC-3 line (Figure 4D, Appendix A). 

### 2.3. Evaluation of Insect Resistance and Agronomic Performance in the Field

Five transgenic lines above were field tested for their resistance to striped stem borers by artificial and natural infection combined and leaffolders by natural infection. The representative phenotype was shown in Figure 5. Wild-type Nipponbare suffered serious damage caused by the pests, while the RRC lines were nearly unaffected. The statistical results of insect damage are shown in Table 1. It was discovered that 68.03% of tillers of the wild-type Nipponbare were severely damaged by striped stem borers, showing dead hearts or white heads, while only slight damage (2.88% at the highest) was observed in these five transgenic lines. Regarding the damage caused by leaffolders, the number of leaves damaged per plant was 0–0.37, while that of the negative controls was 24 (Table 1). These data thus demonstrated that the RRC transgenic lines were highly resistant to striped stem borers and leaffolders under field conditions.

At last, three of the five single-copy transgenic lines, RRC-1, -2, and -3, were selected for agronomic performance evaluation. The data collected from the field experiment for agronomic performance are shown in Table 2. There were two lines, i.e., RRC-1 and -2 showed no significant differences from the wild-type, indicating that RRC-1 and -2 were the ideal line for developing transgenic rice against lepidopteran pests. All the results above followed the original intention of designing transgenic rice transformants with null expression in the endosperm and high expression Bt protein level in insect-attacked parts.

## 3. Discussion

In this study, we took a strategy to fuse a foreign gene with *OsrbcS*, which served as a carrier under the control of its native promoter to sequester its expression exclusively in green tissue. The Cry1Ab/Cry1Ac-rbcS recombinant protein retained the toxin activity, conferring high resistance to the insects in the field test even under the harsh conditions where each rice plant was infested with about 80 first-instar larvae of striped stem borers and natural leaffolders, attacking at the filling stage. A similar strategy was proposed in nutrition fortified lysine-rich (LR) transgenic rice, in which the recombination protein composed of a lysine-rich protein (LRP)-encoding region derived from the winged bean and the glutelin (Gt1) of rice was driven by the GT1 promoter. The safety assessment test via animal feeding indicated that no adverse effects were observed in rats fed transgenic rice compared to non-transgenic rice [29,30]. Fusing foreign genes with endogenous genes may be an attractive alternative to sustaining an optimal abundance in an appropriate organ.

Using an rbcS transit peptide coding sequence to import foreign protein into chloroplast has been reported in many insect-resistant transgenic rice [31,32,33]. However, using the full-length *OsrbcS* as a chaperone has not been reported yet. According to our practice, the fusion construct also exhibited strong resistance to striped stem borers and leaffolders in field experiments. The protein amount assessed by ELISA in green leaf varied between 1.80 and 11.50 μg/g, and was almost 100% effective against striped stem borers and leaffolders. Compared to the previous similar work, our *OsrbcS* fusion construct showed a higher content of Bt protein in the rice leaves. The evidence was listed as follows: The resistant gene, *cry1C**, driven by the rice *rbcS* promoter, was introduced into Zhonghua 11 (*Oryza sativa* L. ssp. *japonica*) by Agrobacterium-mediated transformation. The cry1C*** protein level was only from 0.87 to 3.13 µg g^−1^ in the leaves [21]. The *Cry1Ab* gene was conjugated to the rice *rbcS* promoter to express the Bt toxin in specific tissues of transgenic plants. The Cry1Ab protein level was merely 1.66 to 3.31 µg g^−1^ in the leaves [22]. A novel *cry2AX1* gene consisting of a sequence of the *cry2Aa* and *cry2Ac* genes driven by the rice *rbcS* promoter was introduced into a rice cultivar, ASD16. The expression of the Cry2AX1 protein in transgenic rice plants was only from 0.00595 to 0.12240 µg g^−1^ of fresh leaf tissues [18]. A fusion protein of Cry1Ac and Cry1I-Like driven by a green tissue-specific promoter was introduced into a rice cultivar, Xiushui-134. The expression of the fusion protein in transgenic rice plants was merely 1.050 to 1.510 µg g^−1^ of fresh leaf tissue [34]. Even when compared to the rice *Actin I* promoter-driven control, T51-1, all five RRC transgenic lines are higher than the control in the leaves (Figure 4B). Meanwhile, the protein amount assessed by ELISA in the endosperm ranged between 0.000117 and 0.001173 μg g^−1^. The highest amount is lower than the minimum line of 0.0026 μg g^−1^, as reported by Ye et al. [21]. In addition, three lines in our study showed no declining trend of the Cry1Ab/Cry1Ac protein concentration between the tillering and filling stages in the leaves. However, this declining trend was reported in previous studies [21]. The OsrbcS fusion protein seemed more stable than the unfused form. It is worth emphasizing again that the expression level of the Cry1Ab/Cry1Ac protein is much higher than our previous study [23] and the other transformants with *OsrbcS* unfused studies [21,22], since the high-level expression of the Cry1Ab/Cry1Ac protein in leaves and stems is an effective way to delay the evolution of pest resistance.

Here, we tried to explain why the OsrbcS fusion protein seemed more stable than the unfused form. Firstly, OsrbcS may facilitate the recombinant protein translocating to the chloroplast. The chloroplast is a key hub in converting solar energy into chemical energy to sustain life on earth. The number of chloroplast-targeted nuclear-encoded proteins largely exceeds the number of chloroplast-encoded proteins. The proteome of today’s chloroplasts consists of ∼3000 proteins, most of which are nucleus-encoded and posttranslationally imported into the organelle [35,36]. Accordingly, we concluded that chloroplasts provide an in-built remarkable capacity to allow foreign protein expression.

Moreover, chloroplast lacks the gene silencing modification that precludes sustainable expression over generations [37]. Secondly, it is well accepted that transgenic proteins suffer degradation more rapidly in free cytosol in comparison to their presence in chloroplasts [38]. A fusion protein between the target protein and rbcS was linked with rbcL to make an L8S8 hexadecameric complex structure [39]. rbcS, which is known to be stably present in the form of a protein complex in a chloroplast, may increase the expression of a target protein in plant cells, thereby highly increasing resistance to protein degradation [40]. 

The over-expression of insect-resistant proteins might lead to severe negative effects and abnormal morphology to the host [41,42]. This problem can be circumvented by targeting the insect-resistant protein to the chloroplast. The outer and inner envelope membranes of chloroplast encapsulate the heterologous expression limited to the compartment, without affecting the biology of the rest of the cell [43,44]. Regarding the recalcitrant features for the manipulation of cereal crop plastid transformation [45,46,47], our OsrbcS fusion strategy is not only considered a potential avenue for the expression of the foreign protein to develop null expression in endosperm with higher expression in insect-damaged parts of GM rice, but can also be used as a model to regulate the appropriate expression of other GOI by using endogenous genes fused in GM crops.

## 4. Materials and Methods

### 4.1. Plant Materials and Plasmid Construction

Rice cultivar *Oryza sativa* ssp. *japonica* cv. Nipponbare was used as a recipient for all transformations included in this study. Fragments were cloned into MCS (multiple cloning sites) in the T-DNA of the commercially available binary vector pSB130 via the restriction-ligation cloning method [23]. pSB130 is a standard binary vector, which contains two T-DNA regions and multiple cloning sites (MCS having restriction sites of *Hin*d III, *Sal* I, *Xba* I, *Bam*H I, *Sma* I, *Kpn* I, *Sac* I, and *Eco*R I), and is used to load the target fragments in T-DNA1, while the *hygromycin B phosphotransferase* (*HPT*) gene in T-DNA2 functioned as a transformation selectable marker. The restriction sites of T-DNA1 used to construct the pRY, pRRY, and pRRC plasmids are shown in Figure 1. The expression cassette was assembled in three steps. Firstly, the *Nos* terminator was introduced into the MCS in T-DNA1. Secondly, the GOI (*eYFP* or *Cry1Ab/Cry1Ac* fusion gene) was ligated with the *Nos* terminator. Thirdly, the promoter *pOsrbcS* or *pOsrbcS*:*rbcS* was ligated with the GOI. More specifically, the *pOsrbcS* sequence (3179 bp) was obtained from the genomic DNA of wild-type Nipponbare by PCR amplification with forward 5′-CCCAAGCTTCACTTAAATTTTGGTGACAGG-3′ and reverse 5′-ACGCGTCGACCTCTGCAGCTCACCAAGCTC-3′ primers (restriction sites are underlined). The *pOsrbcS*:*rbcS* sequence (3807 bp) was obtained from the genomic DNA of wild-type Nipponbare by PCR amplification with forward 5′-CCCAAGCTTCACTTAAATTTTGGTGACAGG-3′ and reverse 5′-ACGCGTCGACGTTGCCACCAGACTCCTCGCA-3′ primers (restriction sites are underlined). The *Cry1Ab/Cry1Ac* fusion gene, consisting of the 1,344 bp encoding the N terminus of *Cry1Ab* and 486 bp encoding the C terminus of *Cry1Ac*, was derived from our lab’s previous work by PCR amplification with forward 5′-ACGCGTCGACATGGAC AACAACTGCAGGCC-3′ and reverse 5′-CGGGGTACCTTATTCAGCCTCGAGTGTTG-3′ primers (restriction sites are underlined) [11,23]. A fusion of two toxin genes was used to assure the durability and sustainability of pest resistance.

### 4.2. RNA Extraction and qRT-PCR Analysis 

Total RNA was isolated from fresh plant tissues using TRIzol reagent (Takara, Japan). Each sample was a mixture of at least 20 plants. According to the manufacturer’s instructions, the first-strand cDNA was generated using the Perfect Real Time Primescript RT reagent (TaKaRa, Japan). Real-time quantitative PCR was performed on an optical 96-well plate with a LightCycler 96 Real-Time PCR System (Roche, Switzerland) using SYBR Premix Ex Taq (TaKaRa, Japan), with three technical replicates. The PCR thermal cycling protocol was as follows: 95 °C for 10 s, followed by 40 cycles at 95 °C for 5 s, and 60 °C again for 30 s. The Actin gene was used as the internal reference to assay the relative expression levels of *eYFP*. The gene-specific primers used for qRT-PCR are listed in Appendix A.

### 4.3. Tissue Localization of pOsrbcS:eYFP and pOsrbcS:OsrbcS-eYFP Line

Seedlings were grown in *Yoshida* solution for approximately 2 weeks in a growth chamber [48] (temperature 25 °C; humidity 75%; light ≥ 20,000 lux; 12 h of light and 12 h of the night). The leaves and roots of the young seedlings at the 2–3 leaf stage and the cross-section of the newly germinated seeds derived from transgenic *eYFP* lines were prepared in this study. The fluorescence signals were visualized under confocal microscopy (Zeiss LSM710, Germany ).

### 4.4. Quantitative Estimation of Cry1Ab/Cry1Ac Protein Expressed in Transgenic Plants

A modified procedure using the ELISA kit AP003 CRBS (EnviroLogix, Portland) was implemented to detect the expression level of the Cry1Ab/Cry1Ac protein. Approximately 20 mg of fresh sample from the leaf and stem at the tillering and filling stages, and endosperm at the mature stage, were collected and ground into powder in liquid nitrogen. The powder was then suspended in an extraction buffer according to the 20 mg powder/500 μL extraction buffer proportion and diluted to an appropriate concentration using dilution buffer. The enzyme-linking reaction was performed according to the manufacturer’s instructions. The final concentrations of Cry1Ab/Cry1Ac in the samples (μg·g^−1^ tissue) were calculated based on the standard curve.

### 4.5. Southern Blot

Southern blot analysis was conducted to screen the copy number, and 10 µg of genomic DNA from each sample was digested with *Hin*d III, separated on a 1% agarose gel, and then transferred to a nylon membrane using 20 × SSC solution followed by standard capillary transfer protocol (GM Healthcare, UK). The probes were prepared from a PCR-amplified fragment of *Cry1Ab/Cry1Ac* (Appendix A), which was labeled with DIG-dUTP using a PCR DIG Probe Synthesis Kit (Roche, Switzerland). The transferred DNA was cross-linked by a UV crosslinker at 1200 mJ for 1 min. Hybridization, membrane washing, and subsequent chemical staining with BCIP/NBT (Roche, Switzerland) were conducted as described [49].

### 4.6. Field Tests for Insect Resistance and Agronomic Performance

All the transgenic lines and controls (wild-type Nipponbare) for insect resistance and agronomic traits were planted in the experimental farm of Zhejiang University in Hangzhou, China. The seeds were sown in a seedling bed in early June 2018, and the seedlings were transplanted to the experimental fields approximately 25 days later. The field layout followed a randomized complete block design with three replications. Each plot consisted of 20 plants in 2 rows with distances of 20.0 cm between plants within a row, 15.0 cm between rows, and 40.0 cm between plots. The experimental plots were bordered by three rows of non-transgenic Nipponbare rice plants. In the verification test carried out, no chemical insecticides targeted against lepidopteran pests were applied throughout the growth period. Therefore, natural infestations of leaffolders were obtained. The reaction of the plants to the natural infestation of leaffolders was scored 5 to 7 days after peak damage appeared. Leaves with visible scrapes and tillers with visibly scraped leaves were scored as damaged leaves and tillers, respectively. Both natural and manual infestations of striped stem borers were used. For manual infestations under field conditions, each rice plant was infested with 80 first-instar larvae of striped stem borers at the late booting stage. The insect eggs were bought from China Taobao (https://item.taobao.com/item.htm?spm=a1z10.3-c-s.w4002-24362899300.10.42566b3cA2jvBM&id=698368036197), accessed on 1 September 2018, and hatched in a plant growth chamber at 25 °C and 75% humidity under a photoperiod of 14 h light/10 h dark for 3–5 days to incubate the first-instar larvae. Damage symptoms were checked 7 to 15 days after infestation. Dead hearts and white heads caused by striped stem borers were counted together to calculate the infection rate. 

To test the agronomic performance of the transgenic plants in the field, the plot was designed and used as above, although chemical pesticides were applied throughout the growth period for crop protection. At maturity, the plant height was measured in the field, and all plants from each plot (15 individual plants per line were divided into 3 repetitions) were collected to measure the panicle length, number of panicles per plant, number of grains per panicle, seed-set rate, 1000-grain weight, and yield per plant.

## Figures and Tables

**Figure 1 ijms-24-09045-f001:**
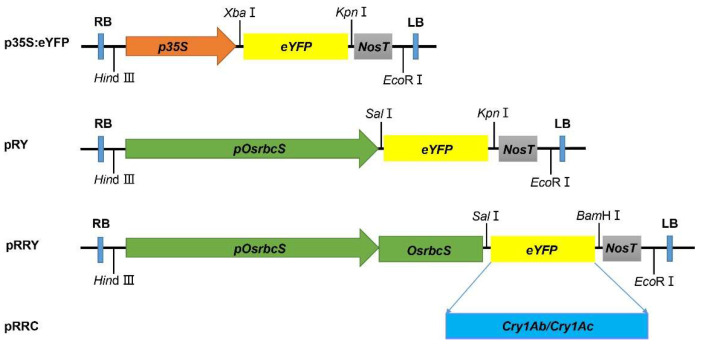
Schematic diagram of T-DNA region of the four transformation constructs in this study: p35S:eYFP; pRY; pRRY; pRRC. All constructs were terminated by the nopaline synthase (*Nos*) terminator. LB: left border of T-DNA region. RB: right border of T-DNA region.

**Figure 2 ijms-24-09045-f002:**
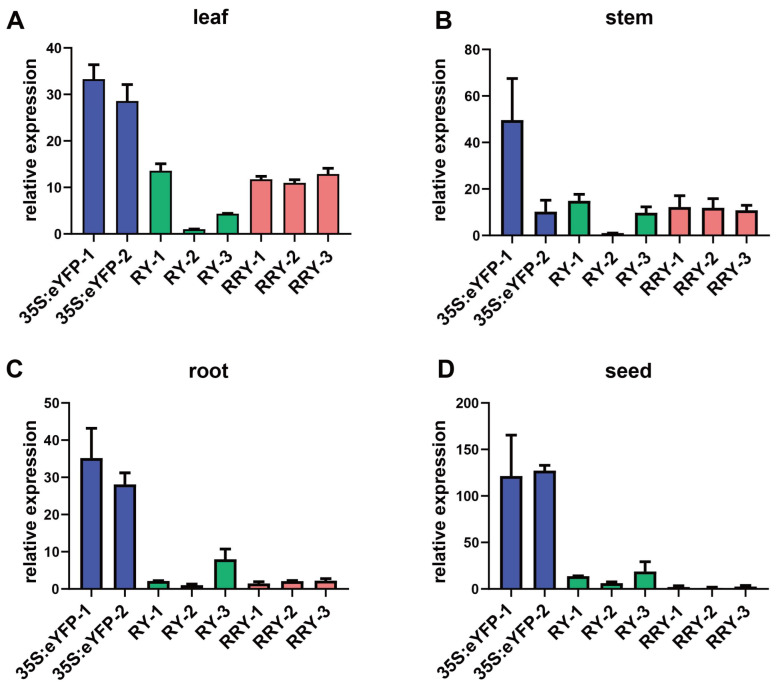
Quantification of the *eYFP* at the mRNA level in different tissues of transgenic rice plants. Two lines for p35S:eYFP and three lines for RY and RRY, respectively. The data are presented as the mean of three biological replicates with error bars indicating the standard error. (**A**) Leaf, (**B**) stem, (**C**) root, (**D**) seed.

**Figure 3 ijms-24-09045-f003:**
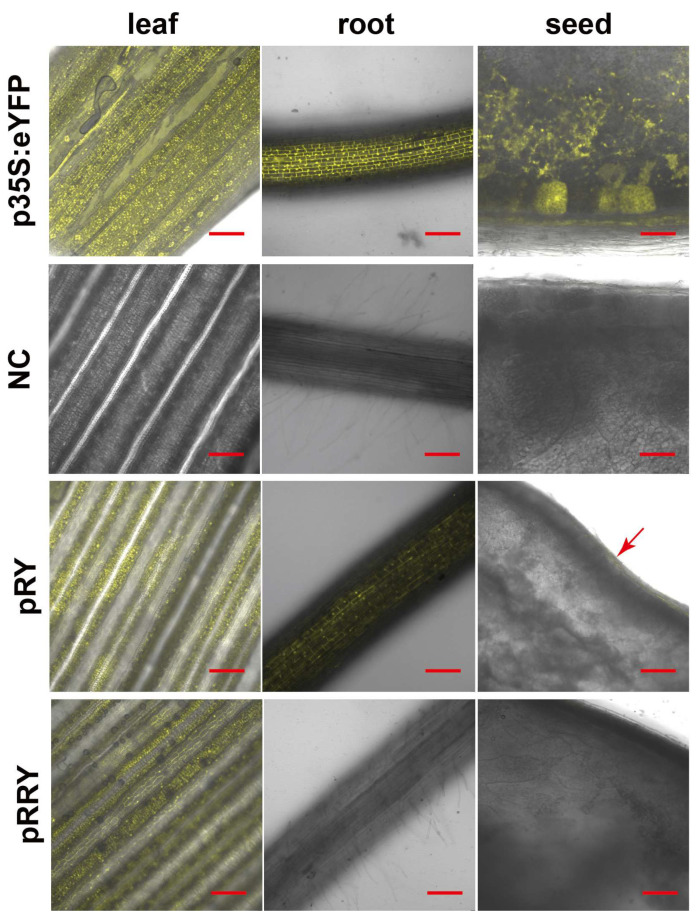
Visualizing eYFP fluorescence in leaf, root, and seed of RY; RRY transgenic rice plants by confocal laser scanning microscopy. p35S-eYFP lines were used as positive controls. Arrow indicated the visible eYFP signal in the aleurone cells of seeds. Wild-type Nipponbare was used as negative control (NC). Bar = 100 μm.

**Figure 4 ijms-24-09045-f004:**
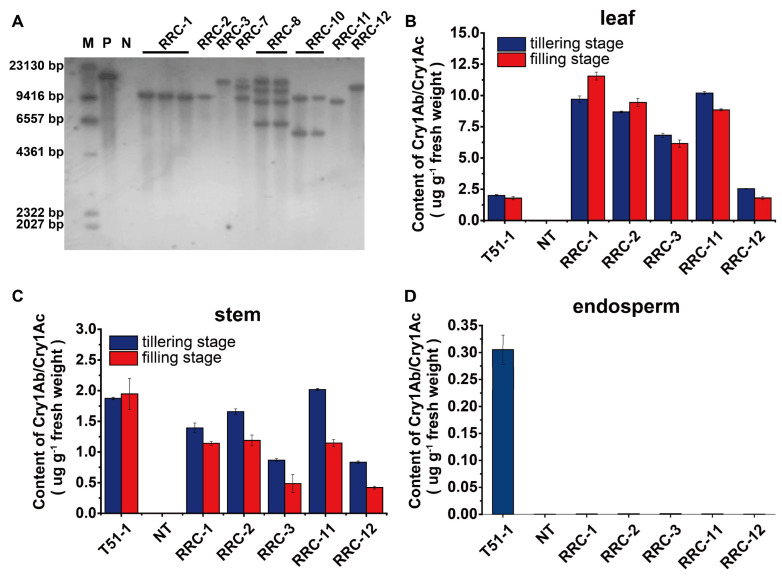
Southern blot analysis and Cry1Ab/Cry1Ac protein assay of RRC lines. (**A**) Genomic DNA extracted from the rice fresh leaves was digested with *Hin*d III and hybridized with a Cry1Ab/Cry1Ac probe; M: DNA marker; P: Plasmid of pRRC; N: Non-transgenic wild-type Nipponbare; Eight independent RRC transformants without significant phenotypic changes were selected from the total 12 independent lines to conduct Southern blot. The underline indicated the same RRC transformants. (**B**) Quantitative expression analysis of Cry1Ab/Cry1Ac protein in the leaves at tillering and filling stage. (**C**) Quantitative expression analysis of Cry1Ab/Cry1Ac protein in the stems at tillering and filling stage. (**D**) Quantitative expression analysis of Cry1Ab/Cry1Ac protein in endosperm at the mature stage. Error bars indicate SD based on three biological replications and two technological replications. T51-1 and non-transgenic wild-type Nipponbare were used as the positive and negative control, respectively.

**Figure 5 ijms-24-09045-f005:**
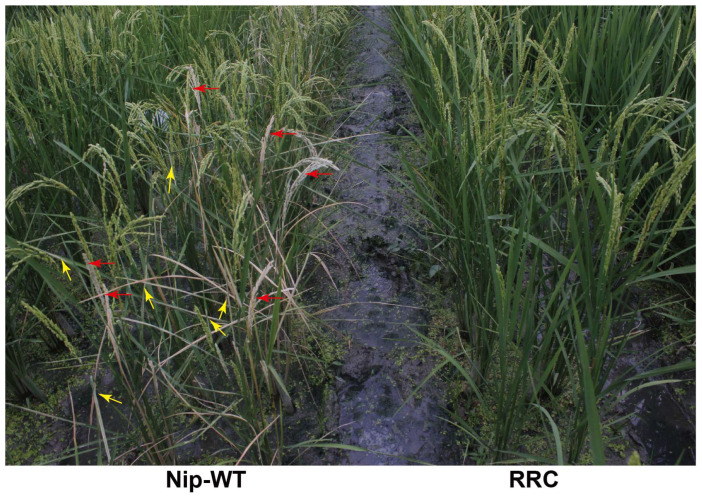
Field performance of RRC transgenic lines against an artificial infestation of striped stem borers and natural infection combined, and natural infestation of leaffolders (red and yellow arrows indicate white heads and fold leaves that were damaged by striped stem borers and leaffolders, respectively). Wild-type Nipponbare was used as the negative control.

**Table 1 ijms-24-09045-t001:** Resistance evaluation of RRC lines and the wild-type Nipponbare against striped stem borers and leaffolders under field conditions (2018, Hangzhou, China).

Line	Damaged by Stem Borers Number of Tillers Affected (%)	Damaged by Leaffolders Number of Damaged Leaves per Plant
Nip-WT	68.03 ± 4.27	24.00 ± 7.05
RRC-1	2.39 ± 2.12 **	0.17 ± 0.21 **
RRC-2	1.28 ± 1.28 **	0.13 ± 0.12 **
RRC-3	2.88 ± 2.66 **	0.06 ± 0.11 **
RRC-11	0.46 ± 0.79 **	0.00 ± 0.00 **
RRC-12	1.08 ± 1.86 **	0.37 ± 0.15 **

All data were measured from 10 randomly sampled plants per test material per replication (3 replications) and were analyzed by Tukey’s multiple comparisons test using the GraphPad Prism 8 software package. Values are given as the mean ± standard deviation (SD). Nip-WT (wild-type Nipponbare) as the negative control; all of the other lines were compared with the Nip-WT. ** meant significantly different from the control at *p* < 0.01.

**Table 2 ijms-24-09045-t002:** Agronomic traits of RRC lines and wild-type Nipponbare under field conditions (2018, Hangzhou, China).

Line	Plant Height (cm)	Panicles per Plant	Panicle Length (cm)	Grains per Panicle	Grains per Plant	Seed-Set Rate (%)	1000-Grain Weight (g)	Yield per Plant (g)
Nip-WT	96.83 ± 1.17	11.53 ± 0.23	17.13 ± 0.62	83.43 ± 2.40	961.93 ± 14.44	87.68 ± 0.76	25.75 ± 0.05	24.77 ± 0.36
RRC-1	96.30 ± 0.52	10.93 ± 0.64	17.80 ± 1.23	87.23 ± 2.78	952.6 ± 31.20	89.67 ± 0.86	25.73 ± 0.13	24.5 ± 0.91
RRC-2	94.57 ± 1.02 *	11.87 ± 0.76	16.99 ± 0.30	81.21 ± 3.06	963.27 ± 63.97	89.8 ± 0.89	25.76 ± 0.30	24.79 ± 1.43
RRC-3	93.8 ± 0.40 **	11.93 ± 1.03	17.53 ± 0.69	27.93 ± 1.44 **	332.33 ± 12.35 **	26.59 ± 1.95 **	25.81 ± 0.12	8.56 ± 0.29 **

All data were measured from 5 randomly sampled plants per test material per replication (3 replications) and analyzed by Tukey’s multiple comparisons test using the GraphPad Prism 8 software package. Values were given as the mean ± standard deviation (SD). Nip-WT was the control and all of the other lines were compared with it. * and ** meant significantly different from the control at *p* < 0.05 and *p* < 0.01, respectively.

## Data Availability

Data will be made available upon request.

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
