# Peer review of "Achieving High Expression of Cry in Green Tissues and Negligible Expression in Endosperm Simultaneously via rbcS Gene Fusion Strategy in Rice"

_ijms, 2023, doi:10.3390/ijms24109045_

Round 1
Reviewer 1 Report
The manuscript describes a novel way in which the expression of the Cry was restricted in green tissues and not detected in endosperm. The seed is used for food consumption and absence of the transgenic material in the seeds is a great achievement for genetic engineering of plants. It is a great scientific work!
Minor comments:
- I suggest slight modification of the title: Achieving high expression of Cry in green tissues and no expression in endosperm simultaneously via rbcS gene fusion strategy in rice
- Figure 4A. Define what the numbers 1 to 12 indicate for clarity
Minor editorial correction may be needed. Example: Line 277. Correct to '...prepared in this study'
Author Response
Dear reviewer,
Thanks for your careful review and good suggestions, we have made the point-to-point response and advised the MS according, more details are shown in the attached file.

Reviewer 2 Report
The authors report on achieving high expression of a Bt Cry1Ab/Cry1Ac in green tissues with low expression in endosperm via the use of a rbcS gene fusion in rice. The results are impressive, and this reviewer appreciates including agronomic values (including yield) along with insect control when greatly increasing expression levels of the target GOI.
General comments
1. The authors use a number of descriptors in their presentation of their results that really does not inform the reader: For example "high", "barely detectable", "high dose", etc. These descriptors can still be used as long as the actual values to support these descriptors are included. Please include actual values
2. The Materials and Methods need improving primarily for the construction of the constructs (including the Cry1Ab/Cry1Ac gene) and the field trials
Specific comments
Line 20: What does “almost none” mean?
Line 25: 1.8-11.5 ppm is really not that high. Compared to what? Please provide values and not just a citation.
Line 26: What does “barely detectable” mean?
Line 27: The authors state that the toxicity was “high dose”. "High dose" has a specific meaning when it comes to insecticidal traits and resistance/durability. "High dose" is defined as 25 times the LC99 and therefore cannot be determined using a field trial where the highest value one could get would be 100% mortality. Please rephrase
Lines 61-62: The plants are not less potent, the insects have just developed resistance. Please rephrase
Line 91: What does “high” amount mean?
Line 225: Not a high dose. Please see comments above for line 27. Please rephrase.
Materials and Methods:
Where is the description of how they made their constructs? Note that this is the primary basis for this MS and therefore should be explained in detail. Where did they get the genes/promoters, etc. from? Please describe the Cry1Ab/Cry1Ac fusion protein and why they chose this GOI versus either Cry1Ab or Cry1Ac?
Lines 299: Field tests: Where were the tests conducted. Where did the insects for the artificial infest come from? How were they reared?
Fig 2. What are the values for the Y-axis? Why relative expression? Please provide actual expression levels
The quality of English is okay, but could use a little improvement. But the quality of English was more than sufficient to this reviewer to understand the MS
Author Response

(The authors gave the same response as above.)

Round 2
Reviewer 2 Report
The authors have adequately responded to this reviewer's comments. However, one comment that really needs to be addressed is in the title:
In the title, the authors have changed "free in endosperm" to "no expression in endosperm". Note this reviewer apologizes for not commenting on "free in endosperm" in the previous review. First of all, even in the abstract on line 27, the authors state "barely detectable" expression in endosperm, and in Results and Discussion actually provide values of expression in endosperm. Finally, in science, one almost never should state "no". It is just that you may not detect it. However, in this MS, the authors could detect expression in endosperm, just negligible amounts of it. So this reviewer would suggest changing "no expression in endosperm" to "negligible expression in endosperm.
Author Response
Dear reviewer,
Thanks for your careful review, we have changed "no expression in endosperm" to "negligible expression in endosperm" as you suggested. We believe that "negligible" is more accurate and precise than "no" in a science paper. Thanks again.